Three new species of Talaromyces sect. Talaromyces discovered in China

Ren Xin-Tong 1
Li Saifei 2
Ruan Yongming 1 394745201@qq.com
http://orcid.org/0000-0001-6774-6999 Wang Long 3 wl_dgk@sina.com
1 College of Life Sciences, Zhejiang Normal University , Jinhua, Zhejiang , China
2 Technology Development and Transfer Center, Institute of Microbiology, Chinese Academy of Sciences , Beijing , China
3 State Key Laboratory of Mycology, Institute of Microbiology, Chinese Academy of Sciences , Beijing , China
Mora-Montes Héctor
Electronic publication date: 2024 Oct 11
Publication date: 2024
Volume: 12
Electronic Location ID: e18253
Received 2024 May 1; Accepted 2024 Sep 16
Copyright: © 2024 Ren et al.
Copyright year: 2024
Copyright holder: Ren et al.
License: This is an open access article distributed under the terms of the Creative Commons Attribution License, which permits unrestricted use, distribution, reproduction and adaptation in any medium and for any purpose provided that it is properly attributed. For attribution, the original author(s), title, publication source (PeerJ) and either DOI or URL of the article must be cited.
License URL: https://creativecommons.org/licenses/by/4.0/

Keywords: Molds, New taxa, Polyphasic taxonomy, Soil fungi, Trichocomaceae

Funding: National Natural Science Foundation of China U20A20101, 31750001 National Project on Scientific Groundwork, Ministry of Science and Technology of China 2019FY100700 This work was supported by the National Natural Science Foundation of China (No. U20A20101, No. 31750001) and the National Project on Scientific Groundwork, Ministry of Science and Technology of China (No. 2019FY100700). The funders had no role in study design, data collection and analysis, decision to publish, or preparation of the manuscript.

==============================
Background

Talaromyces species play an important role in the nutrient cycle in natural ecosystems, degradation of vegetal biomass in industries and the implications in medicine. However, the species diversity of this genus is still far from fully understood.

Methods

The polyphasic taxonomic approach integrating morphological comparisons and molecular phylogenetic analyses based on BenA, CaM, Rpb2 and ITS sequences was used to propose three new Talaromyces species.

Results

Three new species of sect. Talaromyces isolated from soil are proposed, namely, T. disparis (ex-type AS3.26221), T. funiformis (ex-type AS3.26220) and T. jianfengicus (ex-type AS3.26253). T. disparis is unique in low growth rate, velvety texture, limited to moderate sporulation, biverticillate, monoverticillate and irregular penicilli bearing a portion of abnormally large globose conidia, it has no close relatives in phylogeny. Being a member of T. pinophilus complex, T. funiformis produces mycelial funicles on Czapek yeast autolysate agar (CYA), 5% malt extract agar (MEA) and yeast extract (YES), sparse sporulation on Czapek agar (Cz), CYA, MEA and YES while abundant on oatmeal agar (OA), bearing appressed biverticillate penicilli and globose to pyriform conida with smooth to finely rough walls. T. jianfengicus belongs to T. verruculosus complex, is characterized by velvety colony texture with moderate to abundant elm-green conidia en masse, producing biverticillate penicilli, globose conidia with verrucose walls.

Conclusion

It is now a common practice in establishing new species of Aspergillus, Penicillium and Talaromyces based on morphological characters and phylogenetic analyses of BenA, CaM, Rpb2 and ITS sequences. The proposal of the three novelties of Talaromyces in this article is not only supported by their morphological distinctiveness, but also confirmed by the phylogenetic analyses of the concatenated BenA-CaM-Rpb2 and BenA-CaM-ITS, as well as the individual BenA, CaM, Rpb2 and ITS sequence matrices.

Introduction

Talaromyces species are common fungi inhabiting various terrestrial, aquatic (freshwater and marine) and atmospheric environments, which play an important role either in the nutrient cycle of natural ecosystems or the sustainable development of society, for example, producing lignocellulolytic enzymes in plant biomass degradation, synthesizing bioactive secondary metabolites with medical importance, or as the biological control agents in agriculture, etc. But some species are opportunistic pathogens causing talaromycosis in humans and animals, such as T. marneffei (e.g., Yilmaz et al., 2014).

Talaromyces species are often encountered and readily recognized on isolating media due to their yellow, orange or reddish colors in the mycelium and/or substratum. However, in contrast to this superficial recognition, species in this genus are less distinguishable from each other by morphological characters in most cases, especially by micro-morphological characters. The molecular approach has to be used in the identification of these fungi. On the basis of four genetic loci, i.e., β-tubulin gene (BenA), calmodulin gene (CaM), DNA-dependent RNA polymerase II second largest subunit gene (Rpb2) and the nuclear rDNA ITS1-5.8S-ITS2 (ITS), and the integration with morphological characters, about 208 species have been reported in Talaromyces. The genus are currently resolved into eight sections, i.e., sections Bacillispori, Helici, Islandici, Purpurei, Subinflati, Talaromyces, Tenues and Trachyspermi (e.g., Samson et al., 2011; Yilmaz et al., 2014; Sun et al., 2020; Houbraken et al., 2020; Lacey et al., 2023; Liu et al., 2023; Nguyen & Lee, 2023; Špetík et al., 2023; Zang et al., 2023; Mo et al., 2024; Visagie et al., 2024; Okubo, Itagaki & Hirose, 2024). Sect. Talaromyces is the largest section which now includes about 90 species (Houbraken et al., 2020; Lacey et al., 2023; Liu et al., 2023; Nguyen & Lee, 2023; Visagie et al., 2024) (Table 1).

Table 1 The ninety species reported in sect.

Talaromyces, their ex-types and genetic markers, and the three novelties proposed in this study.

Species	Ex-types	Genetic markes	
ITS	Ben A	CaM	Rpb2	
T. aculeatus	NRRL 2129 = CBS 289.48	KF741995	KF741929	KF741975	KM023271	
T. adpressus	CGMCC 3.18211 = CBS 140620	KU866657	KU866844	KU866741	KU867001	
T. alveolaris	CBS 142379	LT558969	LT559086	LT795596	LT795597 (antisense)	
T. amazonensis	CBS 140373 = IBT 23215	KX011509	KX011490	KX011502	MN969186	
T. amestolkiae	CBS 132696 = DTO 179F5	JX315660	JX315623	KF741937	JX315698	
T. angelicae	KACC 46611	KF183638	KF183640	KJ885259	KX961275	
T. annesophieae	CBS 142939	MF574592	MF590098	MF590104	MN969199	
T. apiculatus	CBS 312.59 = FRR 635	JN899375	KF741916	KF741950	KM023287	
T. argentinensis	NRRL 28750	MH793045	MH792917	MH792981	MH793108	
T. aspriconidius	CBS 141835 = DTO 340-F8	MN864274	MN863343	MN863320	MN863332	
T. atkinsoniae	BRIP 72528a	OP059084	OP087524	N/A	OP087523	
T. aurantiacus	CBS 314.59 = NRRL 3398	JN899380	KF741917	KF741951	KX961285	
T. aureolinus	AS3.15865	MK837953	MK837937	MK837945	MK837961	
T. australis	CBS 137102 = IBT 14256	KF741991	KF741922	KF741971	KX961284	
T. bannicus	AS3.15862	MK837955	MK837939	MK837947	MK837963	
T. beijingensis	CGMCC 3.18200 = CBS 140617	KU866649	KU866837	KU866733	KU866993	
T. brevis	CBS 118436 = DTO 004-D8	MN864269	MN863338	MN863315	MN863328	
T. calidicanius	CBS 112002	JN899319	HQ156944	KF741934	KM023311	
T. californicus	NRRL 58168	MH793056	MH792928	MH792992	MH793119	
T. cavernicola	URM 8448	ON862935	OP672383	OP290543	OP290515	
T. cnidii	KACC 46617	KF183639	KF183641	KJ885266	KM023299	
T. coprophilus	CBS 142756	LT899794	LT898319	LT899776	LT899812 (antisense)	
T. cucurbitiradicus	ACCC 39155 = CGMCC 3.26140	KY053254	KY053228	KY053246	OR242024 (CN090C2)*	
T. derxii	CBS 412.89 = NHL 2981	JN899327	JX494306	KF741959	KM023282	
T. dimorphus	AS3.15692 = NN072337	KY007095	KY007111	KY007103	KY112593	
T. domesticus	NRRL 58121	MH793055	MH792927	MH792991	MH793118	
T. duclauxii	CBS 322.48 = NRRL 1030	JN899342	JX091384	KF741955	JN121491	
T. echinulatus	CNUFC HB1206	OR462362	OR507571	OR608367	OR591610	
T. euchlorocarpius	DTO 176I3 = CBM PF1203	AB176617	KJ865733	KJ885271	KM023303	
T. flavovirens	CBS 102801 = IBT 27044	JN899392	JX091376	KF741933	KX961283	
T. flavus	CBS 310.38 NRRL 2098	JN899360	JX494302	KF741949	JF417426	
T. francoae	CBS 113134 = IBT 23221	KX011510	KX011489	KX011501	MN969188	
T. funiculosus	CBS 272.86 = IMI 193019	JN899377	JX091383	KF741945	KM023293	
T. fuscoviridis	CBS 193.69 = IBT 14846	KF741979	KF741912	KF741942	MN969156	
T. fusiformis	CGMCC 3.18210 = CBS 140637	KU866656	KU866843	KU866740	KU867000	
T. galapagensis	NRRL 13068 = CBS 751.74	JN899358	JX091388	KF741966	MH793105	
T. ginkgonis	CGMCC 3.20698	OL638158	OL689844	OL689846	OL689848	
T. haitouensis	AS3.16101	MZ045695	MZ054634	MZ054637	MZ054631	
T. indigoticus	CBS 100534 = IBT 17590	JN899331	JX494308	KF741931	KX961278	
T. intermedius	CBS 152.65 = IMI 100874	JN899332	JX091387	KJ885290	KX961282	
T. kabodanensis	CBS 139564 = DTO 204-F2	KP851981	KP851986	KP851995	MN969190	
T. johnpittii	BRIP 72504a = MST-FP2594	OP712677	OP712647	OP712645	OP712646	
T. kendrickii	CBS 136666 = IBT 13593	KF741987	KF741921	KF741967	MN969158	
T. lentulus	AS3.15689	KY007088	KY007104	KY007096	KY112586	
T. liani	CBS 225.66 = NRRL 3380	JN899395	JX091380	KJ885257	MH793100	
T. louisianensis	NRRL 35823	MH793052	MH792924	MH792988	MH793115	
T. macrosporus	CBS 317.63 = FRR 404	JN899333	JX091382	KF741952	KM023292	
T. mae	AS3.15690	KY007090	KY007106	KY007098	KY112588	
T. malicola	NRRL 3724	MH909513	MH909406	MH909459	MH909567	
T. mangshanicus	AS3.18013	KX447531	KX447530	KX447528	KX447527	
T. marneffei	CBS 388.87 = IMI 068794ii	JN899344	JX091389	KF741958	KM023283	
T. muroii	CBS 756.96 = PF 1153	MN431394	KJ865727	KJ885274	KX961276	
T. mycothecae	CBS 142494	MF278326	LT855561	LT855564	LT855567	
T. nanjingensis	JP-NJ4 = M 2012167	MW130720	MW147759	MW147760	MW147762	
T. neofusisporus	AS3.15415 = CBS 140623	KP765385	KP765381	KP765383	MN969165	
T. oumae-annae	CBS 138208 = DTO 269-E8	KJ775720	KJ775213	KJ775425	KX961281	
T. panamensis	CBS 128.89 = IMI 297546	JN899362	HQ156948	KF741936	KM023284	
T. penicillioides	AS3.15822	MK837956	MK837940	MK837948	MK837964	
T. pinophilus	CBS 631.66 = IMI 114933	JN899382	JX091381	KF741964	KM023291	
T. pratensis	NRRL 62170	MH793075	MH792948	MH793012	MH793139	
T. primulinus	CBS 321.48 = NRRL 1074	JN899317	JX494305	KF741954	KM023294	
T. pseudofuniculosus	CBS 143041	LT899796	LT898323	LT899778	LT899814 (antisense)	
T. purgamentosus	CBS 113145	KX011504	KX011487	KX011500	MN969189	
T. purpureogenus	CBS 286.36 = IMI 091926	JN899372	JX315639	KF741947	JX315709	
T. qii	AS3.15414 = CBS 139515	KP765384	KP765380	KP765382	MN969164	
T. rapidus	CBS 142382 = UTHSC DI 16-148	LT558970	LT559087	LT795600	LT795601	
T. ruber	CBS 132704 = DTO 193H6	JX315662	JX315629	KF741938	JX315700	
T. rubicundus	CBS 342.59 = NRRL 3400	JN899384	JX494309	KF741956	KM023296	
T. rufus	CBS 141834 = CGMCC 3.13203	MN864272	MN863341	MN863318	MN863331	
T. sayulitensis	CBS 138204 = DTO 245H1	KJ775713	KJ775206	KJ775422	MH793141
(NRRL 62185)*	
T. shilinensis	CGMCC 3.20699	OL638159	OL689845	OL689847	OL689849	
T. siamensis	CBS 475.88 = IMI 323204	JN899385	JX091379	KF741960	KM023279	
T. soli	NRRL 62165	MH793074	MH792947	MH793011	MH793138	
T. sparsus	AS3.16003	MT077182	MT083924	MT083925	MT083926	
T. stellenboschiensis	CBS 135665 = IBT 32631	JX091471	JX091605	JX140683	MN969157	
T. stipitatus	CBS 375.48 = NRRL 1006	JN899348	KM111288	KF741957	KM023280	
T. stollii	CBS 408.93	JX315674	JX315633	JX315646	JX315712	
T. striatoconidium	CBS 550.89 = DTO418-H4	MN431418	MN969441	MN969360	MT156347	
T. thailandensis	CBS 133147 = KUFC 3399	JX898041	JX494294	KF741940	KM023307	
T. tumuli	NRRL 6013	MH793071	MH792944	MH793008	MH793135	
T. veerkampii	CBS 500.78 = IBT 14845	KF741984	KF741918	KF741961	KX961279	
T. verruculosus	NRRL 1050 = CBS 388.48	KF741994	KF741928	KF741974	KM023306	
T. versatilis	IMI 134755 = CBS 140377	KC962111	KC992270	MN969319	MN969161	
T. virens	CGMCC 3.25207	ON563152	ON231297	ON470840	ON470841	
T. viridis	CBS 114.72 = NRRL 5575	AF285782	JX494310	KF741935	JN121430	
T. viridulus	CBS 252.87 = FRR 1863	JN899314	JX091385	KF741943	JF417422	
T. wushanicus	CGMCC 3.20481	MZ356356	MZ361347	MZ361354	MZ361361	
T. xishaensis	CGMCC 3.17995	KU644580	KU644581	KU644582	MZ361364	
T. yunnanensis	KUMCC 18-0208	MT152339	MT161683	MT178251	ON703690 (JGT1-103)*	
T. zhenhaiensis	AS3.16102	MZ045697	MZ054636	MZ054639	MZ054633	
T. disparis sp. nov.	AS3.26221 T	PP544888	PP566271	PP566276	PP555175	
T. funiformis sp. nov.	AS3.26220 T	PP544886	PP566269	PP566274	PP555173	
	AS3.26225	PP544887	PP566270	PP566275	PP555174	
T. jianfengicus sp. nov.	AS3.26253 T	PP544889	PP566272	PP566277	PP555176	
	JFL34-5	PP544890	PP566273	PP566278	PP555177	
T. assiutensis (outgroup)	CBS 147.78	JN899323	KJ865720	KJ885260_	KM023305	
Note:

* The sequences of the ex-types are unavailable in GenBank, those of other strains are used instead. The proposed new species and their strains as well as the GenBank numbers of four genetic markers are in boldface.

In a survey of Talaromyces species in China, we discovered five distinctive Talaromyces strains and propose here three new species represented by them belonging to sect. Talaromyces, namely, T. disparis sp. nov., T. funiformis sp. nov. and T. jianfengicus sp. nov.

Materials and Methods

Isolation of fungi

Talaromyces strains were isolated from soil samples collected from Beijing, Hainan Province and Hebei Province using the method described by Malloch (1981), with dichloran rose Bengal chloramphenicol agar (DRBC) as the isolation medium. Five distinctive Talaromyces strains were isolated and deposited in China General Microbiological Culture Collection (CGMCC) as AS3.26221 (BWL1-2L), AS3.26224 (JXL1-2), AS3.26225 (SJZ2-4), AS3.26220 (BWL1-2), and AS3.26253 (JFL18-1), and one strain, i.e., AS3.26224 (JXL1-2) was identified as T. qii.

Morphological observation

For examination of macro-morphological characters, the culturing media of Czapek agar (Cz), Czapek yeast autolysate agar (CYA), 5% malt extract agar (MEA), yeast extract sucrose agar (YES) and Oatmeal agar (OA), and the incubation temperature and time were used according to Pitt & Hocking (2009), Samson et al. (2010). Color names of Ridgway (1912) were referenced in describing the colors of conidia en masse, mycelium, exudate and/or soluble pigment. For examination of microscopic characters, the procedure described by Xu et al. (2022) was followed.

Phylogenetic analysis

Isolation of genomic DNA was carried out in accordance to the procedure of Scott et al. (2000). Partial BenA, CaM, Rpb2 and ITS sequences were amplified with primers described by Glass & Donaldson (1995), Wang (2012), Jiang et al. (2018) and White et al. (1990), respectively. PCR reagent mixture formulation and reaction parameters were referenced to Zang et al. (2023). Amplicons of the target loci were purified and sequenced, then proofread according to the method described by Xu et al. (2022). The finished sequences without primer sequences were deposited in GenBank (BWL1-2 = AS3.26220: ITS = PP544886, BenA = PP566269, CaM = PP566274, Rpb2 = PP555173; BWL1-2L = AS3.26221: ITS = PP544888, BenA = PP566271, CaM = PP566276, Rpb2 = PP555175; JFL18-1 = AS3.26253: ITS = PP544889, BenA = PP566272, CaM = PP566277, Rpb2 = PP555176; JFL34-5: ITS = PP544890, BenA = PP566273, CaM = PP566278, Rpb2 = PP555177; SJZ2-4 = AS3.26225: ITS = PP544887, BenA = PP566270, CaM = PP566275, Rpb2 = PP555174; JXL1-2 = AS3.26224: ITS = MZ220767, BenA = MZ220770, CaM = MZ220773, Rpb2 = MZ221212).

Among the 90 species in Table 1, 89 species were selected for the phylogenetic analysis of the concatenated BenA-CaM-Rpb2 sequences except T. atkinsoniae whose CaM sequence is unavailable in GenBank. Our six strains and one unidentified strain URM 8665, were included in this BenA-CaM-Rpb2 sequence matrix, thus there were 96 strains in total. Moreover, to confirm the novelty of strains AS3.26253 and JFL34-5, an analysis of BenA-CaM-ITS sequences of 101 strains was carried out including the above 96 strains and one additional strain of T. oumae-annae, one additional strain of T. stellenboschiensis and three additional strains of T. verruculosus (Visagie et al., 2015). In addition, individual analyses of BenA, CaM and ITS sequences of the above 101 strains, and Rpb2 sequences of the above 96 strains were also conducted. In all the analyses, T. assiutensis of sect. Trachyspermi was selected as the outgroup species.

Sequence datasets of the concatenated and individual loci were aligned with MUSCLE implemented in MEGA 6 (Tamura et al., 2013), the alignments were checked and edited to generate sequence matrices, then analyzed with the Maximum Likelihood (ML) method, in which the bootstrap method was used to test the phylogeny for running 1,000 replications. The substitution model and rates among sites were determined by the tool “Find Best DNA/Protein Models (ML)” of MEGA 6, “K2+G+I” is suitable for all the sequence matrices, in which gaps were treated as partial deletion as suggested by Hall (2013).

Nomenclature

The electronic version of this article in Portable Document Format (PDF) will represent a published work according to the International Code of Nomenclature for algae, fungi, and plants, and hence the new names contained in the electronic version are effectively published under that Code from the electronic edition alone. In addition, new names contained in this work have been submitted to MycoBank from where they will be made available to the Global Names Index. The unique MycoBank number can be resolved and the associated information viewed through any standard web browser by appending the MycoBank number contained in this publication to the prefix “http://www.mycobank.org/MycoTaxo.aspx?Link=T&Rec=”. The online version of this work is archived and available from the following digital repositories: PeerJ, PubMed Central, and CLOCKSS.

Results

Phylogenetic analyses

PCR amplification generated ca. 420, 670, 826 and 560 bp amplicons for BenA, CaM, Rpb2 and ITS, respectively. Matrices of BenA-CaM-Rpb2, BenA-CaM-ITS, BenA, CaM, Rpb2, and ITS sequences consist of 1,277, 1,256, 332, 482, 480 and 442 sites with gaps, respectively. The resulted phylograms from the concatenated and the individual matrices all supported the delineation of three novel species (Fig. 1, Figs. S1–S5).

Figure 1 ML phylogram inferred from the concatenated BenA-CaM-Rpb2 sequences showing the three new species in boldface.

ML phylogram inferred from the concatenated BenA-CaM-Rpb2 sequences. Bootstrap percentages over 70% derived from 1,000 replicates are indicated at the nodes. New species are indicated in boldface. Bar = 0.05 substitutions per nucleotide position.

Strain AS3.26221 forms a unique clade with no closely related members in either the phylogams based on the BenA-CaM-Rpb2 and BenA-CaM-ITS (Fig. 1, Fig. S1) or those based on individual sequence matrices (Figs. S2–S5), thus, the name T. disparis sp. nov. is proposed for it. Although it is in a clade with T. intermedius, T. viridis, T. panamensis in the BenA-CaM-Rpb2 phylogenetic tree, and in a clade with T. panamensis in the BenA-CaM-ITS tree, there is no bootstrap support. In the Rpb2 phylogram, it is basal to the clade consisting of T. aculeatus, T. echinulatus, T. francoae, T. bannicus, T. kendrickii, T. mangshanicus and T. penicillioides (Fig. S4), but there is still no bootstrap support. Strains AS3.26220 and AS3.26225 group in the T. pinophilus complex which includes twelve members, i.e., T. adpressus, T. annesophieae, T. cavernicola, T. domesticus, T. lentulus, T. mae, T. malicola, T. pratensis, T. sayulitensis, T. soli, T. pinophilus and T. tumuli in all the phylograms except the ITS phylogram (note: T. domesticus and T. sayulitensis are not in this complex in BenA-CaM-ITS and CaM phylograms). The two strains in this study form a clade related to T. cavernicola and strain URM 8665 with 74% and 71% bootstrap support in BenA-CaM-Rpb2 and BenA-CaM-ITS phylograms, respectively, but no closely related taxa in BenA, CaM and Rpb2 phylograms. Therefore, they may represent a new species and are named here as T. funiformis sp. nov. (Fig. 1, Figs. S1–S4). In the phylogram inferred from BenA-CaM-Rpb2 sequence matrix, strains AS3.26253 and JFL34-5 form one clade in the T. verruculosus complex, closely related to the four members of this complex, i.e., T. johnpittii, T. stellenboschiensis, T. verruculosus, and T. yunnanensis with 100% bootstrap support. Moreover, in the phylograms resulted from BenA-CaM-ITS, BenA, CaM, and Rpb2 sequence matrices with additional T. stellenboschiensis and T. verruculosus strains, they still form the clade closely related to the above four members with 98%, 97%, 86%, 100% bootstrap support, respectively, so these two strains are considered a new species in this complex and are named here as T. jianfengicus sp. nov. In the ITS phylogram, two proposed new taxa, i.e., T. funiformis and T. jianfengicus cannot be discriminated from their close relatives, while T. disparis sp. nov. still forms a solitary clade without any relatives (Fig. S5).

Descriptions of new species (Figs. 2–4)

Figure 2 Morphological characters of T. disparis sp. nov.

Morphological characters of T. disparis AS3.26221 T incubated at 25 °C for 7 days. (A–D) Colonies on CYA, MEA, YES and OA. (E–I) Conidiophores. (I) Conidia. Bar = 10 µm.

Figure 3 Morphological characters of T. funiformis sp. nov.

Morphological characters of T. funiformis AS3.26220 T incubated at 25 °C for 7 days. (A–D) Colonies on CYA, MEA, YES and OA. (E–G) Conidiophores. (H) Conidia. Bar = 10 µm.

Figure 4 Morphological characters of T. jianfengicus sp. nov.

Morphological characters of T. jianfengicus AS3.26253 T incubated at 25 °C for 7 days. (A–D) Colonies on CYA, MEA, YES and OA. (E–G) Conidiophores. (H) Conidia. Bar = 10 µm.

Talaromyces disparis Y. Ruan & L. Wang, sp. nov.

MycoBank No: MB 853638

(Fig. 2)

Etymology. The specific epithet refers to its penicilli in different patterns and conidia in different shapes and dimensions.

Holotype. CHINA. HAINAN: Changjiang, Bawang Ridge Nature Reserve, from soil, 19°7′53"N 109°8′2″E, 1,000 m, April 30, 2019, W.-C. Wang BWL1-2L, ex-type culture AS3.26221 (holotype: HMAS 352912, from the dried culture of AS3.26221 on MEA). GenBank: BenA = PP566271, CaM = PP566276, ITS = PP544888, Rpb2 = PP555175.

Diagnosis. This new taxon is characterized by low growth rate, velvety texture, limited to moderate sporulation; biverticillate, monoverticillate and irregular penicilli, and bearing polymorphic smooth-walled conidia with a portion of abnormally large globose ones.

Description

Colonies 10–11 mm diam on CYA at 25 °C after 7 d, low, plane, margins fimbriate; texture velvety; sporulation moderate, conidia en masse Olive-Buff to Dark Olive-Buff (R. Pl. XL); mycelium white; no exudate and soluble pigment; reverse yellowish white. Colonies 13–14 mm diam on MEA at 25 °C after 7 d, slightly deep, with weak concentric plicates, margins regular; texture velvety; sporulation moderate, conidia en masse Light Cress Green (R. Pl. XXXI); mycelium Green-Yellow (R. Pl. V); exudate and soluble pigment absent; reverse Capucine Buff (R. Pl. III). Colonies 12–13 mm diam on YES at 25 °C after 7 d, low, plane; margins fimbriate; texture velvety; sporulation moderate to limited, conidia en masse Light Olive-Gray (R. Pl. LI), mycelium Vinaceous-Fawn to Deep Olive-Buff or Dark Olive-Buff (R. Pl. XL), white at margins; no exudate and soluble pigment; reverse Vinaceous-Rufous (R. Pl. XIV). Colonies 18–20 mm diam on OA at 25 °C after 7 d, sparse, low, plane, margins regular; texture velvety; sporulation limited in centres, conidia en masse Cress Green (R. Pl. XXXI); mycelium Naphthalene Yellow (R. Pl. XVI) in central areas while white elsewhere; exudate absent or limited, clear; no soluble pigment; reverse yellowish white. Colonies 11–12 mm diam on Cz at 25 °C after 7 d, low, plane, sparse, margins regular; texture velvety; sporulation absent; mycelium Sulphur Yellow (R. Pl. V); no exudate and soluble pigment; reverse yellowish white. On CYA at 37 °C after 7 d, no growth.

Conidiophores rising from surface hyphae; stipes 60–80 (–120) × 3–3.5 μm; penicilli biverticillate, monoverticillate and irregular; metulae (3–) 6–8 per vertical, 10–12 × 3–4 μm; phialides (3–) 4–6 per verticil, ampuliform, 10–12 (–14) × 3–4 μm; conidia ovoid, globose to lemon-shaped, 4–4.5 × 3.5–4 μm, abnormally large globose ones about 5–6 μm, walls thick and smooth, some with connectives at both ends.

Talaromyces funiformis Y. Ruan & L. Wang, sp. nov.

MycoBank No. MB 853639

(Fig. 3)

Etymology. The specific epithet refers to the funiculose appearance on CYA, MEA and YES.

Holotype. CHINA. HAINAN: Changjiang, Bawang Ridge Nature Reserve, from soil, 19°7′53″N 109°8′2″E, 1,000 m, April 30, 2019, W.-C. Wang BWL1-2, ex-type culture AS3.26220 (holotype: HMAS 352911, from the dried culture of AS3.26220 on MEA). GenBank: BenA = MZ220770, CaM = MZ220773, ITS = MZ220767, Rpb2 = MZ221212.

Diagnosis. This new taxon is characterized by producing mycelial funicles on CYA, MEA and YES, sparse sporulation on Cz, CYA, MEA and YES while abundant on OA, low growth rate at 37 °C; appressed biverticillate penicilli and globose to pyriform conida with smooth to finely rough walls.

Description

Colonies 26–28 mm diam on CYA at 25 °C after 7 d, low, plane, margins fimbriate; texture velvety and sparsely funiculose and floccose; sporulation sparse, conidia en masse light Pea Green (R. Pl. XLVII); mycelium Pale Vinaceous-Fawn (R. Pl. XL) in central areas and Light Greenish Yellow (R. Pl. V) at margins; no exudate and soluble pigment; reverse Xanthine Orange (R. Pl. III) in central areas, fading into Pale Orange-Yellow (R. Pl. III) at marginal areas. Colonies 38–40 mm diam on MEA at 25 °C after 7 d, slightly deep, plane, margins regular; texture velvety and densely funiculose and floccose; sporulation moderate to sparse, conidia en masse Pea Green (R. Pl. XLVII); mycelium Light Greenish Yellow (R. Pl. V); exudate and soluble pigment absent; reverse Mars Orange to Orange Rufous (R. Pl. II). Colonies 30–32 mm diam on YES at 25 °C after 7 d, low, irregularly sulcate; margins fimbriate; texture velvety and sparsely funiculose and floccose; sporulation sparse, conidia en masse Olive-Gray (R. Pl. LI); mycelium Pale Salmon Color (R. Pl. XIV) in central areas while Pale Yellow-Orange (R. Pl. III) at margins; exudate limited, brown; no soluble pigment; reverse Xanthine Orange (R. Pl. III), fading into Pale Orange Yellow (R. Pl. III) at marginal areas. Colonies 35–37 mm diam on OA at 25 °C after 7 d, low, plane, margins fimbriate; texture velvety; sporulation abundant, conidia en masse Olive-Gray (R. Pl. LI); mycelium pale yellow; exudate absent or limited, clear; no soluble pigment; reverse Cinnamon Rufous (R. Pl. XIV), fading into light yellow at marginal areas. Colonies 12–14 mm diam on Cz at 25 °C after 7 d, low, plane, margins fimbriate; texture velvety; sporulation sparse, limited in centres, conidia en masse Deep Grayish Olive (R. Pl. XLVI); mycelium white mingled with Pale Flesh Color (R. Pl. XIV); no exudate and soluble pigment; reverse Salmon Buff (R. Pl. XIV). Colonies 10–12 mm diam on CYA at 37 °C, deep, margins regular; texture densely floccose; no sporulation; mycelium white; no exudate and soluble pigment; reverse Salmon Buff to Cinnamon-Buff (R. Pl. XIV).

Conidiophores rising from mycelial funicles; stipes 150–300 × 2.5–3.5 μm, smooth-walled; penicilli biverticillate, appressed; metulae 6–8 per vertical, 10–13 × 2–2.5 μm; phialides 6–8 per verticil, acerose, 10–12 × 2–2.5 μm; conidia globose to pyriform, 2–2.5 μm, walls smooth to finely rough.

Additional strains: CHINA, HEIBEI: Shijiazhuang, Daguo Village, from farm soil, 38°5′9″N 114°25′18″E, 70 m, August 20, 2020, S.-Z. Wei SJZ2-4 = AS3.26225. GenBank: BenA = PP566270, CaM = PP566275, ITS = PP544887, Rpb2 = PP555174.

Talaromyces jianfengicus Y. Ruan & L. Wang, sp. nov.

MycoBank No. MB 853553

(Fig. 4)

Etymology. The specific epithet refers to the locale where the ex-type strain was isolated.

Holotype. CHINA: HAINAN: Ledong, Jianfeng Ridge Nature Reserve, from soil, 18°42′14″N 108°49′33″E, 600 m, April 21, 2019, W.-C. Wang JFJ18-1, ex-type culture AS3.26253 (holotype: HMAS 352913, from the dried culture on MEA). GenBank: BenA = PP566272, CaM = PP566277, ITS = PP544889, Rpb2 = PP555176.

Diagnosis. This new taxon is characterized by velvety colony texture with moderate to abundant elm-green conidia en masse and green-yellow mycelium, biverticillate penicilli, ampuliform phialides and globose conidia with verrucose walls.

Description

Colonies 18–19 mm diam on CYA at 25 °C after 7 d, low, plane, margins fimbriate; texture velvety; sporulation moderate, conidia en masse Light Elm Green (R. Pl. XVII); mycelium Chalcedony Yellow (R. Pl. XVII), white at margins; no exudate and soluble pigment; reverse Pale Orange-Yellow (R. Pl. III). Colonies 29–31 mm diam on MEA at 25 °C after 7 d, low, plane, margins regular; texture velvety; sporulation abundant, conidia en masse Elm Green (R. Pl. XVII); mycelium Chalcedony Yellow (R. Pl. XVII), white at margins; no exudate and soluble pigment; reverse Warm buff (R. Pl. XV). Colonies 34–35 mm diam on YES at 25 °C after 7 d, low, convolute centrally, irregularly sulcate densely; margins regular; texture velvety; sporulation abundant, conidia en masse Yew Green (R. Pl. XXXI); mycelium Pale Greenish Yellow R. Pl. V), white at margins; no exudate and soluble pigment; reverse Ochraceous-Salmon to Warm Buff (R. Pl. XV). Colonies 38–40 mm diam on OA at 25 °C after 7 d, low, plane, margins wide, regular; texture velvety; sporulation abundant, conidia en masse Cerro Green (R. Pl. V); mycelium Pale Viridine Yellow (R. Pl. V), white at margins; exudate limited, clear; no soluble pigment; reverse Chalcedony Yellow (R. Pl. XVII). Colonies 16–18 mm diam on Cz at 25 °C after 7 d, low, plane, margins wide, regular; texture velvety; sporulation limited to moderate, conidia en masse Elm Green (R. Pl. XVII); mycelium Chalcedony Yellow (R. Pl. XVII), white at margins; no exudate and soluble pigment; reverse Capucine Buff (R. Pl. III). Colonies 6–7 mm diam on CYA at 37 °C, deep, margins regular; texture velvety; no sporulation; mycelium white; no exudate and soluble pigment; reverse Pinkish Cinnamon (R. Pl. XXIX).

Conidiophores rising from surface and aerial hyphae; stipes 150–200 × 2.5–3 μm when from surface hyphae, and 10–50 μm long when from aerial hyphae, smooth-walled; penicilli biverticillate; metulae 8–10 per stipe, 8–10 × 3.5–4 μm; phialides 6–8 per metula, ampuliform, 10–12 × 3.5–4 μm; conidia globose, 4–5 μm, walls verrucose.

Additional strains: CHINA: HAINAN: Ledong, Jiangfeng Ridge Nature Reserve, from soil, 18°42′14″N 108°49′33″E, 600 m, April 21, 2019, W.-C. Wang JFJ34-5, GenBank: BenA = PP566273, CaM = PP566278, ITS = PP544889, Rpb2 = PP555176.

Discussion

Sect. Talaromyces is the largest section in Talaromyces, including species that commonly grow fast, produce spreading velvelty to floccose colonies more than 30 mm diam. on MEA, and their penicilli are typically biverticillate, though some produce a portion of monoverticillate, terverticillate (with subterminal branches) or irregular penicilli. Some members of this section grow slowly, e.g., T. bannicus, T. mangshanicus and T. marneffei, or form colonies with synnemata or funicles, such as, T. calidicanius, T. duclauxii, T. funiculosus, T. pinophilus, T. pseudostromaticus, but biverticillate penicilli are usually produced, irrespective of whether their phialides are acerose or ampuliform. Phylogenetically, species in this section are usually distantly related, but some closely related taxa compose species complexes, for instance, T. pinophilus complex includes twelve members, i.e., T. cavernicola, T. pratensis, T. lentulus, T. adpressus, T. tumuli, T. soli, T. malicola, T. pinophilus, T. mae, T. domesticus, T. sayulitensis, T. annesophieae; T. verruculosus complex contains four members, namely, T. johnpittii, T. stellenboschiensis, T. yunnanensis, T. verruculosus; T. liani complex consists of T. nanjingensis, T. liani, T. brevis; T. veerkampii complex is comprised of T. californicus, T. louisianensis, T. veerkampii; and T. funiculosus complex comprises T. cucurbitiradicus, T. funiculosus, T. pseudofuniculosus (e.g., Fig. 1). Members of these species complexes are not easily discriminated from each other by using morphology, especially micro-morphology.

Using the polyphasic taxonomy that integrates morphological and phylogenetic characters, we established a new species based on one single strain, since it is phylogenetically distinctive and its morphological characters are obviously distinguishable from other species. T. disparis is a unique taxon that has no close relatives in Talaromyces based on the phylogenetic analysis (Fig. 1, Figs. S1–S5). Though it is in a clade with T. intermedius, T. viridis and T. panamensis in BenA-CaM-Rpb2 phylogram, no bootstrap support is presented. In morphological characters, it is distinctive in low growth rate at 25 °C (CYA: 10–11 mm, MEA: 13–14 mm, YES: 12–13 mm, Cz: 11–12 mm), produces polymorphic conidiophores with biverticillate, monoverticillate and irregular penicilli, bears polymorphic conidia that are ovoid, globose to lemon-shaped commonly measured 4–4.5 × 3.5–4 μm, while with abnormally large globose ones about 5–6 μm. There are few species in sect. Talaromyces that grow slowly on conventional culturing media at 25 °C, such as T. bannicus, T. mangshanicus and T. viridis (Yilmaz et al., 2014; Wang et al., 2017; Wei, Xu & Wang, 2021). T. viridis is readily distinguished from T. disparis by producing ascomata and abnormal anamorphs with solitary phialides and fusiform to ellipsoidal conidia. Besides the restricted growth, T. bannicus and T. mangshanicus both produce biverticillate, monoverticillate and irregular conidiophores with ampuliform phialides similar to those of T. disparis. The definitive difference between T. disparis and T. mangshanicus is that the conidia of the latter are subglobose to ellipsoidal with echinulate walls, obviously different from those of the new species, which are polymorphic with smooth walls. T. bannicus also bears polymorphic conidia with abnormally large ones similar to those of the new species, but their shapes are commonly pyriform to ellipsoidal with walls conspicuously echinulate to verrucose. In all the phylograms, the new species is distantly separated from T. bannicus and T. mangshanicus (Fig. 1, Figs. S1–S5).

Talaromyces funiformis belongs to the T. pinophilus complex. There are now thirteen members in this species complex, which commonly show floccose and funiculose colony appearance, grow well at 37 °C (except T. annesophieae which does not grow), and produce compact penicilli bearing acerose phialides and globose, subglobose to broadly ellipsoidal conidia with smooth to finely rough walls (Visagie et al., 2014; Crous et al., 2017; Jiang et al., 2018; Peterson & Jurjević, 2019; Alves et al., 2022). In the phylograms based on BenA-CaM-Rpb2 and BenA-CaM-ITS, T. funiformis is weakly related to T. cavernicola and the unidentified strain URM8665 with 74% and 71% bootstrap support, respectively. But according to the analyses based on individual loci, except that the ITS region is unable to distinguish between T. funiformis and T. cavernicola, the BenA, CaM, and Rpb2 phylograms all show that they are unrelated. Also morphologically, the floccose and funiculose colony texture, light greenish yellow mycelium and sparse sporulation on CYA, MEA, YES and Cz show similarity to T. cavernicola, but T. funiformis can be discriminated from T. cavernicola by growing more slowly on CYA, YES and Cz at 25 °C (CYA: 26–28 mm vs. 31–35 mm, YES: 30–32 mm vs. 40–47 mm, Cz: 12–14 mm vs. 18–19 mm) and on CYA at 37 °C (10–12 mm vs. 34–38 mm), while much faster on OA at 25 °C (35–37 mm vs. 24–28 mm diam). Moreover, it shows velvety colony texture and abundant sporulation on OA, where T. cavernicola shows floccose texture with poor sporulation. The low growth rate at 37 °C on CYA (10–12 mm) and velvety texture with abundant sporulation on OA of T. funiformis can be used to distinguish it from the other members in this complex, which commonly grow moderately to fast at 37 °C and show floccose and funiculose texture on OA, e.g., T. pinophilus (24–40 mm), T. domesticus (30–38 mm), T. pratensis (25–30 mm), T. soli (23–28 mm), T. tumuli (21–35 mm), T. adpressus (35–38 mm), T. sayulitensis (32–40 mm), T. lentulus (18–21 mm), T. mae (17–18 mm). The only exception is T. malicola which grows slowly at 37 °C on CYA (9–10 mm) similar to T. funiformis (10–12 mm), but the floccose and funiculose texture on OA and low growth rate on MEA (29–31 mm vs. 38–40 mm) of T. malicola still can be relied on to distinguish it from the new species.

The last new species, T. jianfengicus belongs to T. verruculosus complex including T. johnpittii, T. stellenboschiensis, T. yunnanensis and T. verruculosus. These members commonly grow moderately to fast on MEA, YES and OA (note: uncertain in T. johnpittii whose data are unavailable), produce green-yellow mycelium on CYA, MEA and YES, and moderate to abundant conidia in dark green (elm green) and color en masse on MEA, their penicilli usually broadly biverticillate with amuliform phialides bearing globose conidia with echinulate to verrucose walls. The new species produces similar colony appearance, and conidiophores and conidia in shape and dimension to those of T. stellenboschiensis, T. yunnanensis and T. verruculosus, but it still can be discriminated from them by much slow growth at 25 °C on CYA (18–19 mm vs. 40–45 mm, 35–37 mm, 32–35 mm, respectively) and MEA (29–31 mm vs. 40–42 mm, 50–53 mm, 35–36 mm, respectively), and further distinguished from T. stellenboschiensis and T. verruculosus by restricted growth at 37 °C on CYA (6–7 mm vs. 35–40 mm, 25–26 mm, respectively). There are no growth data on CYA, MEA, YES and OA for T. johnpittii, but the new species and T. johnpittii are obviously different in penicilli, for example, the new species produces more numerous metulae and phialides than T. johnpittii (metulae: 8–10 vs. 2–5, phialides: 6–8 vs. 2–5), and the phialide collulae of T. johnpittii are distinctive in showing periclinal thickened collarettes, while those of the new species are conoidal, which is conventional in Talaromyces (Yilmaz et al., 2014; Visagie et al., 2015; Doilom et al., 2020; Lacey et al., 2023).

It is noteworthy that some species in sect. Talaromyces are opportunistic pathogens for humans and animals. T. marneffei is the only thermally dimorphic species in Talaromyces, causing invasive talaromycosis in immunocompromised patients. It is believed that the infection begins with the inhalation of conidia. However, it is puzzling how this fungus produces enough conidia to reach a threshold density for infection in natural habitats, such as soil, since the growth of T. marneffei on artificial culturing media is very slow and the sporulation is sparse. After inhaled, the infectious conidia are primarily engulfed by alveolar macrophages where they germinate to form yeast cells due to 37 °C and the intracellular environment of macrophages. These yeast cells reprogram metabolism pathways to adapt nutrient limitation and synthesize various kinds of virulence factors to survive the stress in macrophages, for example, heat shock protein (HSP), catalase-peroxidase, superoxide dismutase (SOD), melanin and mannoproteins, etc. Some of these proteins, such as HSP, melanin and mannoproteins are included in extracellular vesicles (EVs) secreted into the cytoplasm of macrophage, which stimulate macrophages to produce various inflammatory factors with antimicrobial activity. On the other hand, macrophages may shelter the pathogen from the fungicidal activity of neutrophils (Ellett et al., 2018; Yang et al., 2021; Pruksaphon et al., 2022, 2023; Wang, Han & Chen, 2023). Thus, the pathogenic mechanism of T. marneffei still remains not fully understood.

Many other species in this section were reported to cause talaromycosis, for instance, T. alveolaris, T. amestolkiae, T. aurantiacus, T. cnidii, T. funiculosus, T. indigoticus, T. kabodanensis, T. pinophilus, T. purpureogenus, T. rapidus, T. ruber, T. stipitatus, and T. stollii (Yilmaz et al., 2014; Guevara-Suarez et al., 2016, 2017; Guo et al., 2021; Bacon et al., 2022; Sharma & Nonzom, 2022), whereas, some of these species may be misidentified or in doubt. For example, isolate DI16-138 was identified as “T. cnidii” by Guevara-Suarez et al. (2016), but using its BenA sequence (LT559077) as the query to BLASTn with “sequences from type material”, we found no T. cnidii sequences in the result, and there are nine nucleotides different from that of the ex-type strain of T. cnidii KACC 46617 (KF183641). Guo et al. (2021) identified an isolate PUMCH_Q056 from clinical specimens as “T. aurantiacus”, but the BenA (MW148866) and Rpb2 (MW122767) sequences of the isolate cannot confirm the identification, in addition, according to Bacon et al. (2022), “T. aurantiacus” was isolated from the central nervous system of a Labrador retriever but the ITS sequence (MZ338025) represents Penicillium canescens in GenBank. Sharma & Nonzom (2022) reported T. stipitatus causing superficial mycosis in India, but based on the ITS sequence (MT994164) of their isolate BHS I, it is supposed to be T. zhenhaiensis (ITS= MZ045697).

Irrespective of those pathogenic members, many species in sect. Talaromyces are potent lignocellulolytic enzyme producers and some species have played an important role in the research and application of degrading vegetal biomass that contains cellulose, hemicelluloses, lignin and pectin. Though Trichoderma reesei has been the dominant fungus in studies and industries of ignocellulolytic enzymes, its genome harbors fewer genes encoding cellulase, hemicellulase, especially β-glucosidase than certain Talaromyces species do (Martinez et al., 2008). For example, as the first Talaromyces species utilized in degrading lignocellulosic biomass, T. pinophilus (as “T. cellulolyticus”) presents higher glucan-hydrolyzing and xylan-hydrolyzing activity in its cellulase mixture than Trichoderma reesei does (Fujii et al., 2009), and the genome of T. pinophilus has much more cellulolytic and hemicellulolytic enzyme genes compared with other cellulase-producing fungi (Li et al., 2017). T. funiculosus is also proved to be a good producer for cellulolytic and hemicellulolytic enzymes with more β-glucosidases than Trichoderma reesei produces (Ogunmolu et al., 2015; Pasari et al., 2023). Studies of Goyari et al. (2015) shows that T. verrucullosus secretes efficient β-glucosidase, endoglucanase and cellobiohydrolase on natural cellulose substances with continuous activity without end product inhibition due to sufficient β-glucosidase. Studies of de Eugenio et al. (2017) and Prieto et al. (2021) showed that the genome of T. amestolkiae encodes more cellulolytic and hemicellulolytic enzymes than that of Trichoderma reesei, and the endoxylanase, β-xylosidase, particularly β-glucosidase produced by T. amestolkiae would be good candidates for lignocellulolytic enzyme cocktails. In industries, for example, the endo-1, 3(4)-beta-glucanase and endo-1, 4-beta-xylanase from T. versatilis have been the feed additive for many kinds of domestic animals (e.g., Llanos et al., 2019). All these facts indicate that species in sect. Talaromyces are promising candidates in enzyme industries for plant biomass degradation.

Besides producing lignocellulolytic enzymes, species of sect. Talaromyces are robust producers of secondary metabolites, such as terpenoids, steroids, alkaloids, polyketides, etc. Compared with other sections in Talaromyces, species of this section account for the vast majority in producing various secondary metabolites. According to Nicoletti, Bellavita & Falanga (2023), eleven species from marine environment, namely, T. aculeatus, T. amestolkiae, T. flavus, T. funiculosus, T. mangshanicus, T. pinophilus, T. purpureogenus, T. stipitatus, T. stollii, T. verruculosus, T. versatilis, are the popular producers for new secondary metabolites as well as those shared by other organisms, a large portion of these compounds have antibacterial, anti-proliferative, anti-inflammatory and antioxidant activities, but many aspects of their biotechnological implications remain unexplored.

Conclusions

Talaromyces species are valuable fungal resource either in research work or industrial application, and the species diversity of this genus still needs much investigation. Though the application of environmental metabarcoding technique is widely used in studies on fungal diversity, isolation of pure cultures, discovering and preserving new species are still the prerequisite for in-depth research and exploitation of these biological resources. In this study, we used the serial dilution method in the isolation of fungi and a polyphasic taxonomic approach integrating the morphological and molecular phylogenetic methods in the discovery and proposal of three Talaromyces new species. As the third-generation sequencing technology developed, numerous fungal genomes have been sequenced and assembled in high quality, but much expert bioinformatic analysis still needs to be conducted, which may provide clues for understanding how these fungi survive in natural habitats, pathways of producing lignocellulolytic enzymes and secondary metabolites, as well as the pathogenicity of those opportunistic pathogens.

Supplemental Information

Supplemental Information 1 The sequence matrices analyzed in this study.

Supplemental Information 2 ML phylogram inferred from the concatenated BenA-CaM-ITS sequences showing the three new species in boldface.

Bootstrap percentages over 70% derived from 1000 replicates are indicated at the nodes. New species are indicated in boldface. Bar = 0.05 substitutions per nucleotide position.

Supplemental Information 3 ML phylogram inferred from partial BenA sequences showing the three new species in boldface.

Bootstrap percentages over 70% derived from 1000 replicates are indicated at the nodes. New species are indicated in boldface. Bar = 0.05 substitutions per nucleotide position.

Supplemental Information 4 ML phylogram inferred from partial CaM sequences showing the three new species in boldface.

Bootstrap percentages over 70% derived from 1000 replicates are indicated at the nodes. New species are indicated in boldface. Bar = 0.05 substitutions per nucleotide position.

Supplemental Information 5 ML phylogram inferred from partial Rpb2 sequences showing the three new species in boldface.

Bootstrap percentages over 70% derived from 1000 replicates are indicated at the nodes. New species are indicated in boldface. Bar = 0.05 substitutions per nucleotide position.

Supplemental Information 6 ML phylogram inferred from partial ITS sequences showing the three new species in boldface.

Bootstrap percentages over 70% derived from 1000 replicates are indicated at the nodes. New species are indicated in boldface. Bar = 0.02 substitutions per nucleotide position.

Additional Information and Declarations

Competing Interests

Author Contributions

Data Availability

New Species Registration

The authors declare that they have no competing interests.

Xin-Tong Ren performed the experiments, analyzed the data, prepared figures and/or tables, and approved the final draft.

Saifei Li performed the experiments, analyzed the data, prepared figures and/or tables, and approved the final draft.

Yongming Ruan conceived and designed the experiments, authored or reviewed drafts of the article, and approved the final draft.

Long Wang conceived and designed the experiments, prepared figures and/or tables, authored or reviewed drafts of the article, and approved the final draft.

The following information was supplied regarding data availability:

The raw data is available in the Supplemental File.

The following information was supplied regarding the registration of a newly described species:

MycoBank: http://www.mycobank.org/mb/853638.

http://www.mycobank.org/mb/853639.

http://www.mycobank.org/mb/853553.

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
