# Peer review of "Three new species of Talaromyces sect. Talaromyces discovered in China"

_PeerJ, doi:10.7717/peerj.18253_

## Round 0.1 · original submission · Minor Revisions

Three experts in this field assessed your manuscript and found the results sound and the methodology described in enough detail. However, some issues need to be addressed, particularly with the English usage. In addition, it is important to strengthen the importance of the study and the main conclusion.

Reviewer 1 ·

Basic reporting

The authors present a paper describing 3 novel Talaromyces species from China. While the paper is overall fairly clear, the use of English can be improved in some sections. I would suggest that it is carefully proof read. I made some suggestions on the paper and below. Please pay particular attention to the formatting of the references as it is not consistent throughout the paper.

I would add a final concluding paragraph. Why is this important? What does it mean?

Experimental design

The experimental design is on par with what is expected with a multigene phylogeny supporting the novel species.

Validity of the findings

Describing a species based on a single strain is discouraged, especially as the phylogenetic placement is not consistent in all the trees. Please provide a strong motivation for describing a single strain species.

Additional comments

Additional notes:

Line 37-45: Reduce the length of the paragraph
Line 45: Rewrite: Section Talaromyces is the largest section and includes about 90 species.
Line 87-94: The sentence construction can be improved on. It is difficult to follow.
Line 102: How was the substitution model determined?
Line 129: Rather than referring to “outgroup position”, use “basal to the clade”
Line 129-130: Rewrite the sentence. Avoid starting a sentence with an adverb. Rather say: In the Rpb2 phylogram, it is basal to the clade consisting of T. aculeatus, T. echinulatus, T. francoae, T. bannicus, T. kendrickii, T. mangshanicus and T. penicillioides.
Line 135: The two strains from this study
Line 135-138: Split the sentence in two.
Line 272: Check the spelling of T. marnefeii,
Line 286-287: Split the sentence in two.
Line 288: Please look at the sentence construction.
Line 298-300: Split the sentence in two.
Line 310-313: Rewrite the sentence. The ITS region is unable to distinguish between T. funiformis and T. cavernicola, however, in the BenA, CaM, and Rpb2 phylograms they are unrelated.
Line :

Annotated reviews are not available for download in order to protect the identity of reviewers who chose to remain anonymous.

·

Basic reporting

In the study, three new species of sect. Talaromyces isolated from soil are proposed, namely, T. disparis (ex-type AS3.26221), T. funiformis (ex-type AS3.26220) and T. jianfengicus (ex-type AS3.26253), based on the polyphasic taxonomic approach. Authors also presented the morphological characters of these Talaromyces strains. The paper was interesting.

Experimental design

Methods described with sufficient detail.

Validity of the findings

Conclusions are well stated.

Additional comments

1. The importance and meaning of the study should be strengthened, especially in the Abstract and Introduction section.
2. There were a lot of fungi in the soil, however, authors only isolated five fungal strains from three soil samples. Authors should explain this. If only five fungal strains isolated, I probably guess many fungi were missing.
3. Some biologic activity trials are suggested since the study seems a little simple.
4. Language should be checked carefully by an English native editor.

Reviewer 3 ·

Basic reporting

No comment.

Experimental design

No comment.

Validity of the findings

The discussion needs further elaboration to make this article more comprehensive.

Additional comments

Overall, this manuscript is well-written, concise, and provides a straightforward explanation of the experiments, making it suitable for publication. However, I have some suggestions regarding the discussion section. The author should further elaborate on the significance of the genus Talaromyces in various aspects, such as its industrial applications and medical importance, particularly due to its ability to grow at 37 degrees Celsius. Comparing it with pathogenic fungi in humans and animals, such as the thermally dimorphic Talaromyces marneffei, would enhance the interest and depth of the discussion. I recommend the following open-access articles for further reference: https://www.mdpi.com/2309-608X/8/11/1183 and https://www.frontiersin.org/journals/immunology/articles/10.3389/fimmu.2023.1192326/full.

---

## Round 0.2 · accepted · Accept

The authors addressed the pending comments and consequently, the manuscript is suitable for publication in this journal.